# Every Step Counts—How Can We Accurately Count Steps with Wearable Sensors During Activities of Daily Living in Individuals with Neurological Conditions?

**DOI:** 10.3390/s25185657

**Published:** 2025-09-11

**Authors:** Florence Crozat, Johannes Pohl, Chris Easthope Awai, Christoph Michael Bauer, Roman Peter Kuster

**Affiliations:** 1Therapy Science Lab, Lake Lucerne Institute, 6354 Vitznau, Switzerland; florence.crozat@alumni.epfl.ch (F.C.); christoph.bauer@llui.org (C.M.B.); 2Data Analytics & Rehabilitation Technology (DART), Lake Lucerne Institute, 6354 Vitznau, Switzerland; johannes.pohl@llui.org (J.P.); chris.awai@llui.org (C.E.A.)

**Keywords:** accelerometer, accuracy, activities of daily living, algorithm development, inertial measurement unit, machine learning, neurological population, population-specific algorithm, step count, validation

## Abstract

**Highlights:**

**What are the main findings?**
In individuals with neurological conditions, steps are most accurately counted with a waist sensor, 0.5–3 Hz filter, 5 s window, and gradient boosting regressor. Sensor location has the largest impact on accuracy, followed by window length, regressor type, and filter range.Algorithms trained on able-bodied data detect only 11–47% of steps taken by individuals with neurological conditions during activities of daily living.

**What is the implication of the main finding?**
Population-specific algorithms are essential for accurate step counting in individuals with neurological conditions.Future algorithm developments should incorporate the sensing and analysis configuration identified in this study.

**Abstract:**

Wearable sensors provide objective, continuous, and non-invasive quantification of physical activity, with step count serving as one of the most intuitive measures. However, significant gait alterations in individuals with neurological conditions limit the accuracy of step-counting algorithms trained on able-bodied individuals. Therefore, this study investigates the accuracy of step counting during activities of daily living (ADL) in a neurological population. Seven individuals with neurological conditions wore seven accelerometers while performing ADL for 30 min. Step events manually annotated from video served as ground truth. An optimal sensing and analysis configuration for machine learning algorithm development (sensor location, filter range, window length, and regressor type) was identified and compared to existing algorithms developed for able-bodied individuals. The most accurate configuration includes a waist-worn sensor, a 0.5–3 Hz bandpass filter, a 5 s window, and gradient boosting regression. The corresponding algorithm showed a significantly lower error rate compared to existing algorithms trained on able-bodied data. Notably, all algorithms undercounted steps. This study identified an optimal sensing and analysis configuration for machine learning-based step counting in a neurological population and highlights the limitations of applying able-bodied-trained algorithms. Future research should focus on developing accurate and robust step-counting algorithms tailored to individuals with neurological conditions.

## 1. Introduction

The importance of physical activity for quality of life, morbidity, and mortality is well documented not only in the general population but also among individuals living with neurological conditions [1,2,3]. However, individuals with neurological conditions are considerably less active than their healthy peers [3,4]. For example, stroke, the leading reason for neurological rehabilitation and single biggest driver of years of life lived with disability [5], is associated with markedly low physical activity levels. Stroke survivors accumulate only about half the number of daily steps compared to the general population [6,7,8,9,10]. This lack of physical activity is clinically relevant: It is a major lifestyle-related risk factor for stroke recurrence and a strong predictor of long-term disability, while reaching adequate physical activity levels after stroke is associated with substantial reductions in serious adverse events [11,12,13].

Wearable motion sensors have become the preferred method for objectively tracking physical activity, overcoming the recall bias and social desirability effects that limit self-reported questionnaires [14,15,16]. Modern wearable devices for remote monitoring typically incorporate a triaxial accelerometer, often combined with a gyroscope, and are commonly worn on the wrist, waist, chest, thigh, ankle, or lower back [17,18,19,20,21,22,23]. From the raw sensor signal, researchers derive a wide range of physical activity outcomes, including time spent at different intensity levels, energy expenditure, step count, activity recognition, and activity counts [17,18,21,22,24,25].

From a therapeutic perspective, step count is a particularly valuable outcome because it provides a tangible, meaningful, and easily interpretable metric representing a fundamental unit of human activity that is widely used [8,22,26,27,28]. Step count is seen among the top outcomes for assessing physical activity frequency/volume by researchers and clinicians [24]. An accurate step counter could be used to monitor a patient’s progress over time—detecting both improvements and deteriorations—and to identify patients in need of a physical activity intervention. Furthermore, individualized real-time feedback from step counters could be a fundamental part of the physical activity intervention itself [29,30]. However, the gait patterns commonly observed in neurological populations—characterized by slow walking speeds, short stride lengths, brief and irregular walking bouts, asymmetric walking, compensatory upper body movements, and the use of walking aids—differ substantially from those of able-bodied populations, for which most available step-counting algorithms are designed [31,32,33,34]. For this reason, there is a need to evaluate how steps can be accurately counted with wearable sensors during activities of daily living (ADL) in individuals with neurological conditions [22].

The primary aim of the present study is to determine the optimal sensing and analysis configuration for a neurological population-specific machine learning algorithm to count steps during ADL. The secondary aim is to compare the accuracy of such an algorithm with algorithms trained on able-bodied data. The focus on ADL separates the present study from the vast majority of studies investigating the accuracy of step counting in neurological populations during standardized clinical assessments, such as the 2 or 6 min walking test [35,36,37,38,39,40].

## 2. Materials and Methods

This experimental, cross-sectional study recruited inpatients from the Cereneo Neurorehabilitation Clinic in Hertenstein, Switzerland. This study was reviewed under the Swiss Federal Human Research Act and deemed not to require ethical approval due to its technical focus. All applicable ethical guidelines including the latest version of the Declaration of Helsinki were followed [41].

### 2.1. Participants

Participants were recruited from the clinic’s inpatient population of neurologically impaired adults. For this purpose, a research assistant visited the patients to explain the study. Each patient was free to consent to the study; participation did not affect their treatment. Eligibility criteria included being at least 18 years old, having the ability to understand written and oral study instructions, and having a Functional Ambulation Category score of ≥2 (walking ability with minimal physical assistance, able to support body weight independently). Medical staff approved participant enrolment. All participants provided informed consent prior to study inclusion, and data were pseudonymized to protect privacy.

### 2.2. Data Collection

Data collection took place between 5 November and 18 December 2024 at the clinic facility. The protocol of the present study was designed to capture short indoor walking bouts during typical ADL—varying in length, shape, and speed. Walking was performed under both single- and dual-task conditions (e.g., walking with a cup of coffee), including stair use when deemed safe by the therapist. Activities included the morning routine (e.g., getting dressed, brushing teeth), household tasks (e.g., carrying and fetching items, preparing food), and leisure activities (e.g., walking to the couch, drinking coffee). The detailed protocol can be found in Appendix A. The entire recording lasted approximately 30 min and was supervised by a trained therapist. To reflect ADL, the recorded activities were not standardized. Participants were free to perform them in their own way; only the order was prescribed.

Participants were video recorded (30 Hz) using a handheld Galaxy S23 Ultra (Samsung, Seoul, Republic of Korea) and wore seven triaxial inertial measurement units (AX6, Axivity, Newcastle, UK) recording at 100 Hz (±8 g). Sensors were placed in accordance with previous research at the wrist (left and right, worn with bracelet), chest (medial, taped, between the shoulder blades), lower back (medial, taped, at the level of thoracic vertebra 7), waist (left, clipped to belt or trousers), thigh (left, taped, mid-thigh), and ankle (left, with bracelet, above the malleolus) [17,18,19,20,21,22,23]. Additional sensors not analyzed here were placed on both hands (taped, middle finger axis), both upper arms (taped, mid-biceps), and the right ankle. To synchronize video with sensor data, the left wrist sensor was tapped at both the start and end of the recording while in view of the camera. Video timestamps were subsequently linearly interpolated to synchronize with the sensor timeline [18].

### 2.3. Ground Truth

An experienced annotator (F.C.) manually labelled steps on the video recording using ELAN 6.8 (The Language Archive, Nijmegen, The Netherlands). Steps as part of walking (defined as transferring weight off from one foot to lift it and subsequently reloading it after displacement in space) and shuffling (defined as transferring weight off from one foot and subsequently reloading it without displacement in space) were labelled, i.e., the moment of reloading was labelled [42]. Data were excluded if the participant’s feet were not visible in the video for more than one second (e.g., toilet visits).

To assess inter-annotator agreement, a second experienced annotator (R.P.K.)—blinded to the first annotator’s labels—independently labelled three participants representing different Functional Ambulation Category scores.

### 2.4. Data Processing

Data were processed in Python (version 3.9.21). The selection of sensing and analysis parameters followed the review of Boukhennoufa et al. (2022) [18]. This included (1) sensor location, (2) filtering the raw acceleration signal, (3) windowing (i.e., segmenting the signal into windows) for feature extraction, and (4) training various regressor types to count step events. In line with this review, this study is limited to a window-based approach for feature extraction, which is often considered the default procedure, although simpler alternatives such as threshold crossing or peak detection exist. Only accelerometer data were used in this study to minimize power consumption and, therefore, facilitate future long-term field monitoring. Moreover, a previous investigation found that including a gyroscope provided no additional benefit to quantify physical activity [43].

#### 2.4.1. Sensor Location

Accelerometer data from all locations (left and right wrist, chest, lower back, waist, thigh, and ankle) were analyzed. Each sensor location was analyzed independently. While combinations of sensor locations may offer additional insights, they were not considered in this study as the increased complexity and burden of managing multiple sensors limit their feasibility for future long-term field applications.

#### 2.4.2. Filter Range

Human gait generates acceleration components predominantly below 5 Hz, with the fundamental walking frequency typically not exceeding 3 Hz [44]. However, rapid limb movements, sudden changes, or impact peaks might push energy into higher frequency bands. To investigate the impact of signal filtering, the signal was processed using three a priori selected 5th-order Butterworth filters with zero-phase (forward–backward) implementation: wide-band (low-pass, 15 Hz cut-off frequency), medium-band (band-pass, 0.2–5 Hz), and narrow-band (band-pass, 0.5–3 Hz).

#### 2.4.3. Window Length

Signal features were calculated using sliding windows with 50% overlap across three window sizes: short (0.5 s), medium (2 s), and long (5 s). Time-domain (e.g., mean, standard deviation, number of peaks) and frequency-domain features (e.g., energy, dominant frequency, magnitude of dominant frequency) for the vertical and anteroposterior axes and correlation-based features (e.g., cross-correlation, autocorrelation) between the axes were computed (full feature list given in Appendix A). The vertical and anteroposterior axes have been found to be most reliable in another patient population [45]. Axes were determined according to the SciKit Digital Health method (vertical: highest mean acceleration, anteroposterior: strongest correlation to vertical axis) to ensure signal processing is independent of coordinate system orientation [46].

#### 2.4.4. Regressor Type

Informed by the literature, five methods were evaluated [18]: Gradient Boosting (GB), k-Nearest Neighbors (kNN), Multilayer Perceptron (MLP), Random Forest (RF), and Support Vector Regression (SVR). All regressor types used the signal features as input and step count as output. To reduce the likelihood of overfitting and obtain realistic estimates of generalizability, a leave-one-subject-out (LOSO) cross-validation scheme was employed with three folds. LOSO is well-suited for wearable sensor data, which typically exhibits high inter-subject and low intra-subject variability. It ensures that each validation fold involves a subject not seen during training, thereby simulating deployment to new users [47]. The three test subjects for each parameter combination were randomly drawn from the sample. Each regressor was trained to minimize the root mean squared error (RMSE).

### 2.5. Data Analysis

#### 2.5.1. Annotator Agreement

Inter-annotator agreement was evaluated using the mean absolute percentage error (MAPE) and Pearson correlation coefficients, calculated for each of the three investigated window lengths (short, medium, long). These metrics compare the number of steps identified by the second annotator to the ground truth established by the first annotator.

#### 2.5.2. Optimal Sensing and Analysis Configuration

To answer the primary research question, a generalized linear model was used to estimate the effect of sensor location (reference: waist), filter range (reference: wide-band), window length (reference: medium), and regressor type (reference: GB) on step-counting accuracy. The outcome variable was the median RMSE scaled to 1 s.

A generalized linear model with a gamma distribution and log-link function was applied to account for the positively skewed RMSE values. Model assumptions were assessed visually and statistically (Kolmogorov–Smirnov test). The short window length condition was excluded due to clear violations of distributional assumptions, including a bimodal distribution with substantially higher RMSE values for the short compared to the other window lengths (see Appendix A). If no parameter level significantly outperformed its reference, the reference was retained for the algorithm comparison. Level of significance was set to 0.05. Due to the log-link function, results are directly reported as effect sizes with 95% confidence intervals, *p*-values, and RMSE values. The predicted RMSE for each parameter level represents the expected RMSE when that level is used in combination with all other parameters set to their reference. Parameter importance was quantified by expressing the RMSE of the worst-performing level relative to that of the best-performing level within each parameter.

#### 2.5.3. Algorithm Comparison

To analyze the secondary research question, the optimal sensing and analysis configuration identified in the previous step was used to train a step-counting algorithm using a LOSO cross-validation approach with seven folds. The performance of this neurologically specific algorithm was subsequently compared to four established algorithms originally developed for able-bodied individuals:Threshold-Crossing Algorithm (TCA) detects a step when the vector magnitude of acceleration exceeds a fixed threshold (here: 0.3 m/s^2^) [48].Continuous Wavelet Transform (CWT) applies a Morlet wavelet transform to the signal, followed by peak detection. The wavelet scale adapts to walking speed, improving robustness to temporal variation [49].SciKit Digital Health (SKDH) is a pre-trained machine learning algorithm designed for lower back sensor data [46].OxWearables (OxW) is a pre-trained machine learning algorithm designed for wrist worn sensors [50].

TCA and CWT were applied to the optimal sensor location identified in step 1; SKDH and OxW were applied to the original sensor location (lower back and wrist, respectively). Algorithm performance was compared using the mean absolute percentage error (MAPE) between predicted and ground-truth step counts. Normality was assessed visually (Q–Q plot, see Appendix A) and statistically (Shapiro–Wilk test). If the assumption of normality was not met, a non-parametric Friedman test was applied. Level of significance was set to 0.05. Results are reported with median and interquartile range.

## 3. Results

This study included seven neurological patients (6 male, 1 female) with stroke (*n* = 4), Parkinson’s disease (*n* = 1), spinal cord injury (*n* = 1), and hydrocephalus (*n* = 1). The mean age was 72.7 ± 10.8 years (range: 57–90 years). All participants were right-side dominant and had Functional Ambulation Category scores of 5 (*n* = 2), 4 (*n* = 3), 3 (*n* = 1), and 2 (*n* = 1). One participant used a walking aid (cane).

The mean recording duration was 34.7 ± 8.8 min (range: 20–46 min), during which participants collected an average of 775 ± 301 steps (range: 443–1156 steps). The measurement of one participant had to be repeated due to video loss caused by a software update of the camera. Inter-annotator agreement was high, with an MAPE of 1.3% and Pearson correlation coefficients exceeding 0.99 for each of the evaluated window lengths (short, medium, long).

### 3.1. Optimal Sensing and Analysis Configuration

After excluding the short window length (see Appendix A), visual inspection of model residuals (Appendix A) suggested good model fit. This was confirmed by the Kolmogorov–Smirnov test (*p* = 0.8275), indicating no violation of distributional assumptions.

Significant variations in step-counting accuracy were observed across all four parameters (Table 1). Sensor location had the largest influence on accuracy: RMSE was 100.2% higher for the worst performing location (right wrist, 0.61) compared to the best (waist, 0.30, *p* ≤ 0.001). Window length had the second largest effect: RMSE was 29.6% higher for the worst performing length (medium, 0.30) compared to the best (long, 0.23, *p* ≤ 0.01). Regressor type had the third largest effect: RMSE was 14.1% higher for the worst performing model (kNN, 0.34) compared to the best (RF, 0.30, *p* ≤ 0.01). Finally, filter range showed the smallest effect: RMSE was 10.9% higher for the worst performing range (wide, 0.30) compared to the best (narrow, 0.27, *p* ≤ 0.01).

Based on these results, the optimal configuration uses the waist sensor, the narrow filter band (0.5–3 Hz), the long window (5 s), and GB.

### 3.2. Algorithm Comparison

Compared to the ground-truth step count, the algorithm trained with the optimal sensing and analysis configuration counted a median of 86.4% of total steps [interquartile range: 10.2%]. In contrast, CWT counted 47.4% [21.2%], SKDH counted 39.5% [23.2%], TCA counted 18.1% [33.6%], and OxW counted 11.2% [21.7%] of total steps.

This corresponds to an MAPE of 13.6% [9.3%] for GB, 52.6% [21.3%] for CWT, 60.5% [23.3%] for SKDH, 81.9% [33.6%] for TCA, and 88.8% [21.7%] for OxW (Figure 1). The Friedman test indicated a significant difference in algorithm accuracy (*p* = 0.002). A non-parametric test was chosen because the Shapiro–Wilk test approached the level of significance (*p* = 0.053), and the Q–Q plot showed substantial deviations from normality (see Appendix A).

## 4. Discussion

This study investigated the accuracy of step-counting algorithms during ADL in individuals with neurological conditions, who typically exhibit altered gait patterns compared to healthy peers. Impaired gait and limited mobility significantly affect functional independence and quality of life [51,52], and step count serves as a clinically meaningful outcome measure to quantify physical activity [24,26,27]. The results of this study demonstrate that step-counting accuracy varies significantly across sensing and analysis parameters. The optimal configuration—a waist sensor combined with a 0.5–3 Hz filter range, a 5 s window, and a gradient boosting regressor—achieved the lowest error rate (mean MAPE of 13.6%). In contrast, the algorithms developed for able-bodied individuals performed markedly worse in this sample (mean MAPE of 52.6–88.8%).

Using step frequency values for stroke survivors during controlled clinical walking assessments (1.66 steps/s during the 6 min walking test [40]), these error rates translate to 0.23 steps/s for the population-specific algorithm and 0.87–1.48 steps/s for the able-bodied-trained algorithms. These values exceed the error margins typically observed in controlled clinical settings (e.g., 2 or 6 min walking test), where step-counting errors range from 0.04 to 0.45 steps/s [36,37,39,40]. However, similar error margins have been observed in other studies conducted outside standardized test conditions (MAPE: 27.0–320.8% or 0.45–5.34 steps/s) [53]. Furthermore, similar error margins are found when using step frequency values observed in the present study (775 steps in 34.7 min or 0.37 steps/s on average): 0.05 steps/s for the population-specific algorithm and 0.19–0.32 steps/s for the able-bodied-trained algorithms. Whether these error margins are sufficient for rehabilitation monitoring remains unclear, as clinical relevance likely depends on disease-specific factors, mobility levels, and total daily step counts. Of note, all algorithms investigated in the present study underestimate the number of steps compared to the ground truth. We speculate that atypical gait patterns (e.g., shuffling with weak acceleration peaks) and isolated single steps outside a gait sequence are potential reasons for this observation and recommend future studies to investigate the misclassification pattern in greater temporal detail.

Sensor location showed the largest effect on step-counting accuracy among the investigated parameters, with the waist yielding a significantly lower RMSE compared to all other locations. Notably, the lower back and ankle sensors also demonstrated acceptable performance, with effect sizes within 10% of the waist sensor (Table 1). This is in line with a previous study that estimated distance walked in stroke survivors and identified the waist and ankle as the most accurate location [54]. While one could expect the ankle to outperform the waist due to its proximity to the ground, the unilateral placement may have limited the ankle’s ability to detect steps of the contralateral foot. This might be particularly relevant during ADL including steps that are not part of bilateral gait patterns, such as brushing teeth or preparing food. The value of sensors mounted on the core, such as the waist and lower back, has previously been shown with respect to gait asymmetries [55]. The observed difference in performance between the waist and lower back was unexpected and warrants further investigation. We pre-selected the waist as the reference because it is the most frequently used sensor location for measuring physical activity [17,18,21] as well as for practical reasons related to wear comfort and donning/doffing. For long-term field monitoring, a waist-worn sensor clipped to the belt or trousers likely offers superior wear comfort compared to a taped sensor on the lower back, which could interfere with backrests during seated activities and might require assistance for donning/doffing. Conversely, from a data perspective, taping a sensor may improve the accuracy of sensor placement and wear compliance, particularly over long wear periods. When combined with previous research of our group, the ankle might be the preferred choice if step counting is to be combined with a tracking of body posture, while the waist and lower back might be the preferred choices if step counting is to be combined with other physical activity outcomes [43]. For the remaining locations, accuracy declined with increasing distance from the feet (thigh to chest to wrist), a trend consistent with expectations based on biomechanical signal attenuation and movement variability at more cranial/distal locations during ADL. However, when considering wearing comfort, the wrist could become a very interesting alternative. The wrist would allow combining step counting with upper extremity usage, a very well researched topic [17,18]. However, the results observed in this study (Table 1, Figure 1) make the wrist the least preferred choice from an accuracy point of view.

Window length was the second most influential factor, with longer windows (5 s) yielding a lower RMSE rate compared to shorter ones (2 s). The shortest window length (0.5 s) had to be excluded from the analysis due to very poor performance on the RMSE (see Appendix A). Window length determines the amount of temporal information captured in each analysis segment (i.e., feature). Very short windows may capture only partial steps, especially in individuals with slow or irregular gait, and are more susceptible to motion artifacts during non-walking activities, making feature extraction less meaningful and stable and, ultimately, negatively impacting step-counting accuracy. In contrast, long windows are more likely to encompass complete or multiple steps, resulting in more robust and representative feature sets. The longest window length investigated in this study yielded the lowest RMSE. As previous research has used even longer windows, up to 10 s [18], future studies should evaluate whether extending window lengths further improves performance during ADL, where many steps may occur outside of structured walking bouts. Conversely, if the goal is to reduce window length to increase temporal resolution for real-time applications and enable the detection of individual steps that are not part of walking bouts, future research should consider alternative feature sets, e.g., without correlation-based features that might be less reliable in very short segments.

Regressor type also influenced performance, with ensemble methods such as GB and RF outperforming simpler models like kNN. This suggests that more complex, non-linear models are better suited to capturing the heterogeneous and irregular movement patterns observed in individuals with neurological conditions and confirms previous research [18]. Although RF yielded a slightly lower RMSE, its performance did not significantly exceed that of GB. In line with our predefined procedure of retaining the reference when no alternative significantly outperforms it, GB was selected for the optimal configuration. Nevertheless, RF demonstrated comparable accuracy and may be considered an equally valid alternative. Ensemble methods aggregate multiple weak learners, enabling them to model subtle, high-dimensional relationships between features and step events, which is particularly valuable when signal characteristics vary between individuals. In contrast, simpler models such as kNN rely heavily on local similarity in feature space and may be more vulnerable to inter-subject variability, which is expected to be pronounced in clinical populations. Interestingly, while neural network models like the MLP offer theoretical flexibility, they did not outperform tree-based methods in this study, potentially due to the limited size of the training data. Our findings support the use of ensemble learning methods in wearable sensor-based gait analysis, particularly in scenarios involving small datasets, which are common in research on clinical populations. Future studies with larger samples should systematically compare ensemble methods with neural network models to better understand their relative strengths.

The filter range, the fourth processing parameter investigated, showed a moderate influence on accuracy. The narrowest band (0.5–3 Hz) yielded the lowest RMSE, likely because it captures the frequency components most relevant to gait while effectively suppressing high-frequency noise and low-frequency drift. The medium band (0.2–5 Hz) is a viable alternative, whereas the wide band (low-pass with a 15 Hz cut-off) does not provide comparable performance.

The algorithm comparison showed that the neurological population-specific step-counting algorithm, trained with the optimal sensing and analysis configuration, achieved the lowest error rate. The able-bodied algorithms reached MAPEs that were 4 to 6.5 times higher. However, it is important to note that this comparison is inherently biased. The population-specific algorithm—despite using LOSO cross-validation—was both trained and validated on the same sample, while the able-bodied algorithms used independent development samples. LOSO cross-validation makes overfitting much more likely and thus limits generalizability of the presented algorithms as compared to independent validation. Consequently, future research should confirm our findings using an independent neurological sample. Such a study should pay close attention to the composition of the sample, including age, diagnosis, Functional Ambulation Category, and use of assistive devices, and, ideally, use an independent sample for validation. Notably, a previous study observed a 6% decrease in performance when moving from LOSO cross-validation in the development sample to validation in an independent sample of healthy participants [56]. A similar or larger reduction in performance should be expected here. To improve the robustness of the population-specific algorithm, we recommend further training on a larger and more diverse neurological cohort prior to independent validation. Interestingly, the ranking of the able-bodied algorithms’ MAPEs in the algorithm comparison (secondary research question, Figure 1) mirrors the ranking of sensor locations in the search for the optimal sensing and analysis configuration (primary research question, Table 1): the waist (CWT) achieved the lowest error, followed by the back (SKDH) and the wrist (OxW) with the substantially largest error. The only exception is the simple TCA algorithm applied to the waist, which had a higher MAPE than SKDH for the back sensor. This aligns with the regressor comparison, where simpler models like kNN and SVR produced the highest error rates.

### Methodological Considerations

A key strength of the present study is its focus on ADL, including typical non-walking tasks such as tooth brushing and food preparation. This contrasts with the vast majority of comparable studies, which rely on standardized clinical gait assessments involving continuous, uninterrupted walking, such as the 2 or 6 min walking tests [35,36,37,38,39,40]. The ADL investigated in this study or by Henderson et al. 2022 are far more representative of patients’ activities in their home environment, strengthening the study’s ecological validity [53]. However, this realism comes at the cost of higher error rates [17,22,36,37,39,40,53,57]. Two key factors contribute to this trade-off. First, the average step frequency in our study was around 0.37 steps/s versus 1.66 steps/s observed in the 6 min walking test [40], reflecting the fragmented, irregular nature of daily movements. Second, recent work has shown that non-stepping tasks are frequently misclassified as steps when algorithms are trained or validated exclusively on structured walking tasks, undermining their generalizability to real-life settings [53]. In this context, it should be noted that this study was limited to indoor ADL performed in a rehabilitation clinic, which may not fully capture the variability encountered in home environments.

This study included a diverse neurological sample (Functional Ambulation Category scores ranging from 2 to 5), with four out of seven patients being stroke survivors. Although the sample is relatively small, this study was sufficiently powered to address the primary research question regarding optimal sensing and analysis configuration, as evident from the significant differences observed for all investigated parameters (Table 1). The sample size in combination with the significance level (0.05) rendered effect sizes of ≥9% statistically significant. Whether smaller effect sizes are clinically meaningful remains a question for future research. We consider the sample size to represent a reasonable balance between the need for statistical power and the ethical responsibility to minimize participant burden. However, the small sample limits the generalizability of the results and precludes subgroup analyses. It was also insufficient to perform a post hoc Wilcoxon test for the algorithm comparison (secondary aim). With seven participants, the smallest possible Wilcoxon *p*-value is 0.0156 (i.e., 2 × 0.5^7^), which cannot reach significance when correcting for multiple comparisons. Therefore, we recommend that the next step involves training an algorithm using the optimal sensing and analysis configuration identified in this study on a larger neurological sample, followed by a validation on an independent dataset. The limited sample size also supported our choice to express the primary results as effect sizes (Table 1) and to constrain the number of signal features to reduce regressor complexity. Future studies with larger samples should consider incorporating more sophisticated features or broader feature sets and employing evidence-based feature selection methods, such as those described previously [58,59]. Additionally, collecting more than 35 min of data or exceeding 775 steps per participant is advisable to further enhance model robustness. Although sensors were placed on both ankles, only the left ankle was analyzed, reflecting our focus on sensor locations most relevant for field-based monitoring (waist and wrists). While no asymmetries were observed in upper-limb movement, lower-limb asymmetries were not explored. Inter-annotator reliability was verified in a small subset of three participants and showed high agreement, indicating that annotation consistency was adequate for the purposes of this study.

The search for optimal sensing and analysis parameters followed a structured approach using generalized linear modelling to obtain results as robust as possible. An alternative would have been to select the single best-performing parameter combination. However, we argue that, in algorithm development, particularly for neurological populations, the primary goal should not be to maximize performance on a specific dataset but rather to prioritize robustness and generalizability. This is especially critical when the intended application involves remote monitoring of ADL. This focus on robustness is also one of the main reasons why the present study incorporated a broad range of ADL not directly related to walking, supporting this study’s ecological validity.

The present study investigated step count as a measure of physical activity volume and frequency [24]. Other gait-related variables, such as gait speed, stride length, or stance duration, may provide even greater clinical insights. Gait speed, for instance, is a strong predictor of functional status, independence, and quality of life and is sometimes referred to as the “sixth vital sign” [22,60]. For assessing other step-based gait characteristics, such as stride length or stance duration, in real-world environments outside standardized clinical assessments, we view accurate step counting as an essential first step. Therefore, we recommend that future algorithm developments for these gait outcomes build on algorithms that can reliably detect steps [61,62]. We fully acknowledge that gait speed has a substantial impact on step detection accuracy [39,40,63,64]. This likely contributes to the poor performance of algorithms developed on able-bodied individuals when applied to neurological populations. Finally, this study did not distinguish between steps taken during walking and those during shuffling. While this distinction may be less critical for general step counting, it could be important in the context of physical activity, where differentiating gait intensity may be relevant.

## 5. Conclusions

This study is among the first structured investigations of how sensor location, filter settings, window length, and algorithm type affect machine learning-based step detection in neurological populations during ADL. The results demonstrate that population-specific algorithms are essential for accurate step counting. The optimal sensing and analysis configuration includes a waist-mounted sensor, a 0.5–3 Hz filter, a 5 s window, and a gradient boosting regressor. The corresponding algorithm detected, on average, 86% of the steps taken by the individuals with neurological conditions, while the able-bodied trained algorithm detected ≤47% of the steps. Notably, all algorithms tended to underestimate step counts relative to the ground truth. This highlights the inherent challenges of step detection in individuals with neurological conditions, particularly during non-walking activities that are common in daily life but poorly represented in traditional gait assessments such as the 6 min walking test. Future studies are needed to evaluate the generalizability of these findings in larger and independent neurological samples.

## Figures and Tables

**Figure 1 sensors-25-05657-f001:**
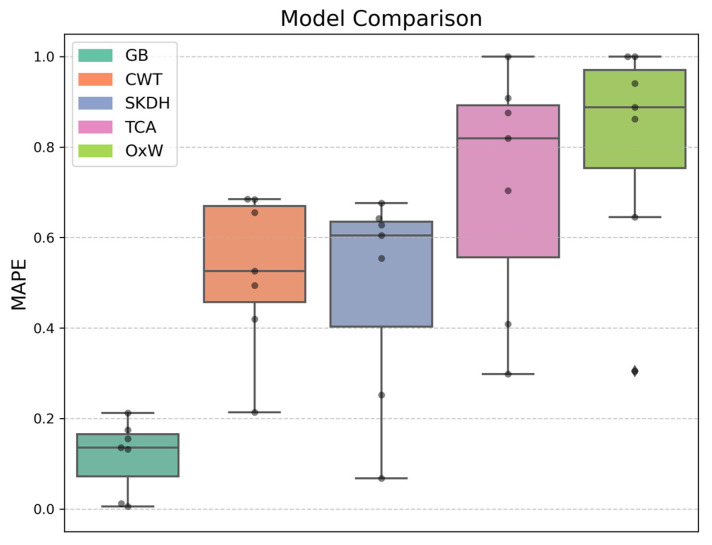
Mean absolute percentage error (MAPE) of the investigated algorithms in n = 7 participants: Gradient Boosting (GB, waist sensor), Continuous Wavelet Transform (CWT, waist sensor), SciKit Digital Health (SKDH, lower back sensor), Threshold-Crossing Algorithm (TCA, waist sensor), and OxWearables (OxW, wrist sensor). Only the GB algorithm was developed in this study using data from individuals with neurological conditions; the remaining algorithms were developed in other studies using data from able-bodied individuals.

**Table 1 sensors-25-05657-t001:** Results of the generalized linear model assessing the effects of sensor location, filter range, window size, and regressor type on the root mean squared error (RMSE) of step counting. Significant differences compared to the reference are indicated with an asterisk (*). Effect sizes and their 95% confidence intervals (95% CI) are reported as exp (β) and 95% CI (exp (β)), respectively, with exp (β) representing the multiplicative change in RMSE relative to the reference. Data for each level is shown relative to their parameter reference, with all other parameters held at their reference.

Parameter	Level	Effect Size	95% CI	*p*-Value	RMSE
Reference (Ref)					0.30
Location (Ref = Waist)	Lower Back *	1.100	[1.002, 1.206]	0.044	0.33
	Ankle *	1.101	[1.004, 1.208]	0.041	0.33
	Thigh *	1.195	[1.089, 1.310]	<0.001	0.36
	Chest *	1.276	[1.164, 1.400]	<0.001	0.39
	Wrist (l) *	1.848	[1.685, 2.027]	<0.001	0.56
	Wrist (r) *	2.002	[1.825, 2.196]	<0.001	0.61
Filter (Ref = wide band)	narrow band *	0.902	[0.849, 0.958]	0.001	0.27
	medium band *	0.911	[0.857, 0.968]	0.003	0.28
Window (Ref = medium)	long *	0.772	[0.734, 0.811]	0.000	0.23
Regressor (Ref = GB)	RF	0.989	[0.914, 1.069]	0.776	0.30
	MLP	1.063	[0.983, 1.150]	0.125	0.32
	SVR *	1.089	[1.007, 1.178]	0.032	0.33
	kNN *	1.128	[1.043, 1.219]	0.003	0.34

Abbreviations: left (l), right (r), Gradient Boosting (GB), Random Forest (RF), Multilayer Perceptron (MLP), Support Vector Regression (SVR), k-Nearest Neighbors (kNN). Note that the short window condition was excluded due to modelling constraints (see Appendix A).

## Data Availability

The raw data supporting the conclusions of this article will be made available by the authors on reasonable request.

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
