# Peer review of "Every Step Counts—How Can We Accurately Count Steps with Wearable Sensors During Activities of Daily Living in Individuals with Neurological Conditions?"

_sensors, 2025, doi:10.3390/s25185657_

Round 1
Reviewer 1 Report
Comments and Suggestions for Authors
This manuscript addresses the use of wearable IMUs and video data to quantify step counts during activities of daily living (ADL) in individuals with neurological impairments. While the topic is important and relevant, the paper requires substantial revisions before it can be considered for publication. The methods are insufficiently described, critical terminology is unclear, and several analytical decisions lack justification. Major clarifications are needed regarding participant characteristics, sensor placement, data collection, and algorithm design, as well as a more rigorous description of step definition and validation.
General Comments
- Authors should avoid using non-standard acronyms. For example, “PA” should be fully spelled out.
- Several methodological descriptions are vague or incomplete, which makes it difficult to evaluate the reproducibility and reliability of the findings.
Methods
Participants
- The neurological impairments and their distribution must be clearly described. Heterogeneity in impairments increases variability in ADL performance, making comparisons with other systems and algorithms difficult.
Data Collection
- The activities performed and their order need to be described in detail.
- It should be clarified what type of “sensor” was used to synchronize video and IMU data.
Step Definition
- The manuscript must provide a formal, quantitative definition of what constitutes a “step.” How did the video labeler decide whether a specific movement (e.g., during turning) was a step?
- The statement “Only accelerometer data were used with respect to power consumption in future long-term field monitoring” is vague and requires clarification.
Sensor Placement and Use
- It is unclear why combinations of sensors were not used, as even simple two-sensor combinations could significantly improve results. Using 2 instead of one sensor will not add significant burden in doing the experiments.
- The rationale for selecting only certain sensor placements needs clarification. For example, IMUs were placed on both wrists but only the left ankle was analyzed. How would this approach affect step counts in individuals with unilateral impairments (e.g., stroke on the left vs. right lower limb)?
Signal Processing
- Filtering range: If acceleration data is expected to be below 5 Hz, why was a cut-off of 15 Hz used? This seems illogical and must be justified.
- Windowing: The use of windowing needs explanation. For a general reader, it is not clear why simple peak counting cannot be used to detect steps. The manuscript should explain the necessity of extracting multiple time- and frequency-domain features.
- Regression: The inputs and outputs for the regressor should be explicitly defined. The rationale for selecting multiple algorithms must be explained, including the pros and cons of each.
Results
- The number of participants appears limited, and variability among them seems large. Would this not compromise the accuracy of the algorithm? Why were more participants not recruited?
- In Table 1, the type of algorithm/regressor used for the first three rows is unclear. For each row (e.g., location), parameters such as window length, filter range, and other settings need to be reported.
Reviewer 2 Report
Comments and Suggestions for Authors
Review sensors-3803573-peer-review-v1
The manuscript titled: Every Step Counts – How can we accurately count steps with wearable sensors during activities of daily living in individuals with neurological conditions?” presents a timely and relevant study on the accuracy of step-counting algorithms using wearable sensors in individuals with neurological conditions. The article is well-structured, technically rigorous, and addresses a meaningful gap in existing literature: the limitations of algorithms trained on able-bodied individuals when applied to a neurologically impaired population. The methodology is solid, featuring a well-defined experimental design, appropriate statistical modeling, and a critical interpretation of the results.
Abstract:
Lines 26–27: The phrase "wearable sensors are a promising tool to quantify physical activity" is vague.
Recommendation: Replace with “Wearable sensors provide objective, continuous, and non-invasive quantification of physical activity.”
Introduction:
The introduction offers strong contextual grounding, clearly outlines the clinical relevance of physical activity tracking post-neurological injury, and identifies a specific gap in current methodologies.
Line 80–81: The sentence “evaluate how can we accurately count steps” is grammatically awkward.
Recommendation: Rephrase to “evaluate how steps can be accurately counted.”
Methods:
The methods are well described and appropriate for the research question. However, two areas would benefit from clarification:
Lines 134–136: Inter-annotator agreement was evaluated using only three participants.
Recommendation: Justify why this limited subset is sufficient to demonstrate reliability, or discuss its limitations.
Lines 137–144: Gyroscope data was excluded from analysis.
Recommendation: Provide a brief justification within the main text (not just via citation), citing power efficiency or prior evidence of limited added value.
Results:
The results are presented and supported by statistical analysis. However, two points need clarification:
Lines 248–250: There appears to be a discrepancy—Gradient Boosting (GB) is presented as both the reference model and the best-performing model, despite Random Forest (RF) having a comparable or slightly better RMSE.
Recommendation: Clarify why GB was selected (e.g., for consistency, robustness, or other practical considerations).
Lines 259–265: The description of Figure 1 omits the number of participants.
Recommendation: Explicitly state the sample size (n = 7) to aid interpretation of variability and error margins.
Discussion:
The discussion effectively interprets the findings and situates them in the broader literature. A few key areas require expansion:
Lines 289–297: All algorithms, including the best-performing one, underestimated step counts.
Recommendation: Offer possible explanations (e.g., misclassification of shuffling or stationary movements) and suggest practical mitigation strategies.
Lines 373–375: While the manuscript acknowledges that leave-one-subject-out (LOSO) cross-validation does not equate to true external validation, it should explicitly address the risk of overfitting.
Recommendation: Recommend future validation using an independent dataset and discuss generalizability limitations.
Lines 443–445: Accurate step detection is described as foundational for gait analysis.
Recommendation: Include a citation to support this assertion and reinforce its clinical significance.
Figures and Tables
- Figure 1 and Table 1 are well-designed and support the key messages.
- Supplementary materials are referenced appropriately.
Citations and References
- References are recent (many from 2022–2025) and relevant.
- Literature coverage is comprehensive.
The manuscript is highly relevant and scientifically sound. Addressing the above issues will strengthen clarity, objectivity, and rigor. Most revisions relate to clarification or minor additions, not major structural changes.
Reviewer 3 Report
Comments and Suggestions for Authors
This study investigated how to accurately count steps in the activities of daily living (ADL) of patients with neurological diseases. The study evaluated the impact of sensor location, filtering range, window length, and regressor type on step detection accuracy, and compared the algorithms developed for specific populations with existing algorithms trained on data from healthy populations. The results are compelling and emphasize the necessity of customizing algorithms for populations with neurological diseases. However, the manuscript requires substantial revisions to enhance methodological clarity, statistical rigor, and the generalizability of conclusions. The following are the primary issues that need to be addressed.
- This study included only seven participants, and the types of neurological disorders were heterogeneous. Although statistically significant effects were reported, the small sample size limited the generalizability of the results. Please discuss this limitation more explicitly and consider performing a stratified analysis of the results by diagnosis or walking ability.
- Although there is a high degree of consistency among annotators, the description of the definition of steps (“dragging steps” and “walking steps”) is vague. Please provide clearer operational definitions and examples to ensure the reproducibility of the study.
- It is reasonable to exclude the 0.5-second window due to poor performance, but its impact and significance on real-time applications should be discussed.
- Comparing algorithms tailored to specific populations (trained and tested on the same sample) with pre-trained algorithms based on healthy populations introduces methodological bias. A more equitable comparison would involve training all algorithms within the same leave-one-out cross-validation (LOSO) framework.
- The article mentions the extracted features, but does not list them in detail. Please provide a complete list of features or a table in the supplementary materials to ensure transparency. In addition, for detailed methods of feature extraction, please refer to the following literature: A new method proposed for realizing human gait pattern recognition: Inspirations for the application of sports and clinical gait analysis (https://doi.org/10.1016/j.gaitpost.2023.10.019).
- Please specify whether the Butterworth filter uses forward-backward (zero phase) filtering or causal filtering, as this will affect signal delay and the possibility of real-time applications.
- Please convert the mean absolute percentage error (MAPE) and root mean square error (RMSE) into clinically interpretable metrics (e.g., steps per minute error) and discuss whether the achieved accuracy is sufficient for rehabilitation monitoring.
Round 2
Reviewer 3 Report
Comments and Suggestions for Authors
All comments have been addressed.